# A Comparative Study of a Potent CNS-Permeable RARβ-Modulator, Ellorarxine, in Neurons, Glia and Microglia Cells In Vitro

**DOI:** 10.3390/ijms26083551

**Published:** 2025-04-10

**Authors:** Yunxi Zhang, Lilie Gailloud, Alexander Shin, Jessica Fewkes, Rosella Pinckney, Andrew Whiting, Paul Chazot

**Affiliations:** 1Department of Biosciences, Durham University, Durham DH1 3LE, UK; yunxi.zhang@durham.ac.uk (Y.Z.); lilie.gailloud.21@ucl.ac.uk (L.G.); alexander.shin@durham.ac.uk (A.S.); jessica.fewkes@durham.ac.uk (J.F.); rosella.a.pinckney@durham.ac.uk (R.P.); 2Department of Pharmacology, University College London, London WC1E 6BT, UK; 3Department of Chemistry, Science Laboratories, Durham University, South Road, Durham DH1 3LE, UK; andy.whiting@durham.ac.uk

**Keywords:** ellorarxine, DC645, NVG0645, retinoid, mitochondrial dysfunction, neuroinflammation, neurodegeneration, autophagy, neuroprotective effects, RARs

## Abstract

Vitamin A (retinol) and its derivatives (retinoids) assume critical roles in neural development, cellular differentiation, axon elongation, programmed cell apoptosis and various fundamental cellular processes. Retinoids function by binding to specific nuclear receptors, such as retinoic acid receptors (RARs) and retinoid X receptors (RXRs), activating specific signalling pathways in the cells. The disruption of the retinoic acid signalling pathway can result in neuroinflammation, oxidative and ER stress and mitochondrial dysfunction and has been implicated in a wide range of neurodegenerative diseases. The present study explored the potential therapeutic application of our innovative CNS-permeable synthetic retinoid, Ellorarxine, for the treatment of neurodegenerative disorders in vitro. An MTT (3-(4,5-dimethylthiazol-2-yl)-2,5-diphenyltetrazolium bromide) tetrazolium assay, lactate dehydrogenase (LDH) assay, enzyme-linked immunosorbent assay (ELISA), immunocytochemistry and immunofluorescence staining were performed. Ellorarxine increased Cyp26 and, selectively, RARβ protein expression in neurons, glia and microglia. Ellorarxine significantly reduced cell death (neurons, glia), increased mitochondrial viability (neurons), modulated cytokine release (microglia), and positively regulated cellular autophagy (neurons, glia, microglia). These results suggest that Ellorarxine is a promising drug candidate that should be further investigated in the treatment of neurodegenerative diseases.

## 1. Introduction

Vitamin A and its derivatives, known as retinoids, are specific modulators for neural differentiation, motor neuron outgrowth and immunology in vertebrates [1,2,3,4]. Retinoids have gained considerable attention in the context of their ability to regulate the gene expression of varieties of encoded enzymes, neurotransmitter transporter proteins and receptors, transcription factors, cell surface receptors and neuropeptide hormones [5]. All-trans retinoic acid (RA), a metabolite of vitamin A, performs physiological functions by binding to and activating RA receptors (RARs) and retinoid X receptors (RXRs), which each have three subtypes (α, β and γ) with several isoforms [6]. RA translocates across the nuclear membrane through RARs, interacts with retinoic acid response elements (RAREs), and participates in the mechanisms of gene regulation [7]. RARs and RXRs exhibit broad expression across nearly all tissues, especially in the brain, although the distribution of each isotype varies [8]. In addition to their genomic effects, retinoids have also important non-genomic effects, mediating homeostatic synaptic plasticity and neurotransmitter release [9,10].

RAR deficits have been proven to be closely related to neurodegenerative diseases. Studies have shown that in vitamin A-deficient rats, the expression of RARα is inhibited, leading to the deposition of amyloid beta (Aβ) peptide in cerebral blood vessels [11]. RARβ and RXRβ/RXRγ mRNA in the hippocampus, when downregulated, causes young animals with VAD to show cognitive decline like that seen for aged animals [12]. RARβ is involved in the neuroprotection of striatal medium spiny neurons (spMSNs), a cell type affected in different neuropsychiatric diseases and particularly susceptible to degeneration in Huntington’s disease (HD) [13]. Retinoid deficiency or mutations in the RARβ and RXRγ genes are associated with the inhibition of spatial learning and memory and the development of depression in animals [14]. RA is often associated with and modulates regions of high neuroplasticity, and in the hippocampus, RA signalling is regulated by the availability of RALDH1 and RALDH2 synthetases and the Cyp26b1 catabolic enzyme [15]. Retinoids are critical for long-term potentiation (LTP) and long-term depression (LTD) mechanisms associated with learning and memory, as well as homeostatic synaptic plasticity [16].

Retinoids play an important role in preventing neuroinflammatory responses to provide neuroprotection, and retinoids can downregulate the expression of cytokines and inflammatory molecules in microglia [17]. Retinoids also regulate the expression of tyrosine hydroxylase, dopamine β-hydroxylase, and dopamine D_2_ receptors [18]. Reduced acetylcholine (ACh) in neurodegenerative diseases has also been implicated in RA-mediated reductions in ChAT production and neuronal cell death [19]. The functional neuroprotective effects of synthetic retinoids in neurodegenerative diseases have been widely studied, and some selective RXR and RAR modulators have been developed as potential drugs for the treatment of neurodegenerative diseases [20,21,22].

In this study, the efficacy of a synthetic retinoid (Nevrargenics’ lead drug, an RAR modulator, Ellorarxine, also known as DC645 and NVG0645) was evaluated on neurodegenerative mechanisms in vitro. RARs are expressed in a variety of cells, including C6, SH-SY5Y and HMC3 cells. The regulation of RAR expression regulates cellular functions [23,24,25]. C6 is an established cell line derived from rat glioma that can differentiate into astrocyte-like cells, express glial fibrillary acidic protein (GFAP) under specific conditions and has been used to culture astrocyte models [26]. SH-SY5Y is a human neuroblastoma cell line that can differentiate into neuron-like cells and can serve as a neuronal model for neurodegenerative diseases [27]. Microglia are related to a series of neurodegenerative diseases (such as attention deficit disorder and Parkinson’s disease) and can have neuroprotective or neurotoxic effects [28]. Herein, the human microglial clone 3 cell line (HMC3) was used as a brain microglial model [29].

## 2. Results

### 2.1. Ellorarxine Upregulates the Expression of Cyp26b1 and RARβ

To investigate the regulation of RAR expression by Ellorarxine, quantitative immunofluorescence was used to detect the expression of Cyp 26B1 and RARs [30].

The results showed that Ellorarxine significantly upregulated the protein expression of Cyp26b1 (Figure 1) (C6: 118%; SH-SY5Y: 47%; HMC3: 36%) and RARβ (C6: 35%; differentiated SH-SY5Y: 71%; HMC3: 46%) but not RARα or RARγ (Figure 2). The regulation of the cell topology of RARα and RARγ by Ellorarxine was more pronounced: Ellorarxine caused RARα in C6 cells to migrate towards one pole of the nucleus (Figure 2A,I, white arrow) and RARγ in differentiated SH-SY5Y neurons to migrate towards the cell membrane (Figure 2D,J, white arrow).

### 2.2. Ellorarxine Pretreatment Alleviates Mitochondrial Dysfunction

To investigate the effect of Ellorarxine on mitochondrial viability, we examined the percentage of mitochondrial viability with or without a 4 h Ellorarxine pretreatment of glia, neurons and microglia under control and stress conditions using an MTT assay. To induce mitochondrial dysfunction, oxidative stress was applied to the cells using 100 µM hydrogen peroxide (H_2_O_2_).

A two-way ANOVA showed a significant difference in the stress condition (*p* < 0.0001) and treatment condition (*p* < 0.0001) in SH-SY5Y cells. The oxidative stress elicited by 100 µM H_2_O_2_ reduced mitochondrial viability by 20–30% in the three cell types, and Ellorarxine had a significant enhancement effect (10% and 17%) on mitochondrial function under both the control and oxidative stress conditions, respectively, in SH-SY5Y neuronal cells (Figure 3A,B).

### 2.3. Ellorarxine Pretreatment Reduced Cell Death

To investigate the effect of Ellorarxine on necrotic cell death, an LDH release assay was performed on C6, SH-SY5Y and HMC3 cells with or without a 4 h Ellorarxine (10 nM) pretreatment. To induce cell death, a final concentration of either 200 µM H_2_O_2_ or 15 μg/mL of LPS was applied.

A two-way ANOVA showed a significant difference in the stress condition (*p* = 0.0004) and treatment condition (*p* = 0.0298) in C6s cells; a significant difference in the stress condition (*p* < 0.0001) and treatment condition (*p* = 0.0019) in SH-SY5Y cells; and no significant difference in HMC3 cells. Ellorarxine had significant cytoprotective effects of 19% and 10% on SH-SY5Y and C6 cells (Figure 4A,B), respectively, in the presence of 200 µM H_2_O_2_. Furthermore, Ellorarxine showed a significant cytoprotective effect (14%) in the presence of 10 μg/mL LPS on SH-SY5Y cells (Figure 4C).

### 2.4. Ellorarxine Pretreatment Modulated Inflammatory Cytokine Release in HMC3 Microglia

The effects of Ellorarxine on neuroinflammation were explored in HMC3 cells with or without a 4 h Ellorarxine (10 nM) pretreatment. To induce inflammation, LPS 10 μg/mL was applied.

The results showed the release of TNF-α (Figure 5A) and IL-6 (Figure 5B) under 10 μg/mL LPS stress (Figure 5A). A two-way ANOVA showed a significant difference in the stress condition (*p* = 0.0034) and no significant different in the treatment condition in TNF-α release and a significant difference in the stress condition (*p* < 0.0001) and treatment condition (*p* < 0.0001) in IL-6 release. Ellorarxine significantly reduced the release of IL-6 by 38.4% (Figure 5B).

### 2.5. Ellorarxine Treatment Regulated Cellular Autophagy

To investigate the regulation of autophagy by Ellorarxine, immunocytochemical staining was used to determine the expression of LC3BII in cells with or without a 4 h Ellorarxine (10 nM) treatment. Immunofluorescence labelling was used to detect the expression of p62 in cells with or without a 4 h Ellorarxine (10 nM) treatment.

A two-way ANOVA showed a significant difference in the stress condition (*p* < 0.0001) and treatment condition (*p* < 0.0001) in C6, SH-SY5Y and HMC3 cells. Ellorarxine treatment significantly elevated the level of LC3B in C6, SH-SY5Y and HMC3 cells in serum-free media (Figure 6D) (C6: 56%; SH-SY5Y: 27%), and downregulated the level of p62 in C6, SH-SY5Y and HMC3 cells under oxidative stress conditions (Figure 7D), therefore showing that Ellorarxine was able to positively regulate cellular autophagy.

## 3. Discussion

This study aimed to evaluate Ellorarxine’s RAR subtype selectivity and the ability of Ellorarxine to ameliorate a range of pathophysiological mechanisms in the three key cell types in the mammalian brain, namely neurons, glia and microglia, relevant to neurodegenerative disease without adverse effects. Studies have shown that RARβ plays an important role in controlling neurotransmission, energy metabolism and transcription [31]. Many of the identified RARβ target genes associated with these pathways have been implicated in various neurodegenerative diseases, such as Alzheimer’s and Parkinson’s diseases [32,33]. Furthermore, studies have shown that the loss of RARβ can lead to mitochondrial dysfunction in mice and that RARβ agonists can prevent mitochondrial failure induced by mitochondrial toxins and reduce mitochondrial fragmentation and cell death [13,34]. Defects in mitochondrial respiratory chain function, oxidative stress, morphology/kinetics, and the calcium-handling capacity can induce neurodegenerative diseases [35,36].

Our results showed that Ellorarxine can significantly alleviate mitochondrial dysfunction in neurons induced by oxidative stress. Under control conditions, Ellorarxine can also significantly improve neuronal mitochondrial function. This may be because Ellorarxine can upregulate the expression of RARβ and activate the neuroprotective effect of RARβ. Indeed, immunofluorescence semi-quantitative analysis results showed that Ellorarxine could significantly upregulate the expression of RARβ in all three cell types without affecting the expression levels of RARα and RARγ. This suggests that Ellorarxine exhibits a level of selectivity for RARβ, acting as an RARβ agonist to enhance the function of RARβ, thereby exerting neuroprotective effects [37,38,39]. This may also be due to the upregulation by Ellorarxine of the expression of Cyp26b1. The Cyp26 family of enzymes (CYP26A1, B1 and C1) includes key proteins that regulate the internal levels of RA in cells, and retinoids are the only substrates of this enzyme family. Cyp26b1 plays an important role in establishing the RA gradient. The RA metabolite 4-oxo-RA produced by Cyp26b1 catabolism was previously shown to be a potent agonist specifically targeting RARβ [15,40,41,42,43,44]. Therefore, the selectivity of Ellorarxine for RARβ and its high potency may be derived from this metabolite produced by its hydroxylation of RA.

In a variety of neurodegenerative diseases, the inflammatory response triggered by xenobiotics, chemicals, beta-amyloids, etc., is driven by inflammatory and pro-inflammatory cytokines and chemokines (TNF-alpha, IL-6, etc.) [45,46]. Microglia exert neuroprotective or neurotoxic effects depending on the intensity of the stimulus and the extent of the inflammatory response. Excessive cytokine release can overactivate microglia, resulting in neurotoxicity [47]. Our results showed that neither TNF nor IL-6 release was stimulated under baseline conditions. Under LPS exposure conditions, Ellorarxine reduced the production of IL-6 by 40% compared with the control group. At the same time, Ellorarxine significantly reduced the neuronal death induced by LPS exposure. These results indicate that Ellorarxine can reduce neuroinflammation and provide neuroprotection. The differential IL-6-inhibitory effect and lack of an effect on TNF contrasts with previous studies in differentiated NC-34 SOD-1 mutant motor neurons, where Ellorarxine inhibited both TNF and IL-6 upon exposure to LPS [48].

In a normal state, damaged organelles and protein aggregates reach the lysosome through endosomal and autophagosomal delivery, where they are digested and recycled through cellular autophagy [49]. In a variety of neurodegenerative diseases, defects occur at different stages of the autophagic pathway, causing neurons to degenerate due to the accumulation of ubiquitinylated protein aggregates [50,51]. In mammals, there are four LC3 isoforms (LC3A, LC3B, LC3B2 and LC3C), which are expressed differently in different tissue cells. LC3B is highly expressed in the brain and endocrine tissues, so we chose to use LC3B, a mammalian homologue of Atg8, with phosphatidylethanolamine as the marker for detection [52]. When autophagy occurs, the LC3I type is modified by ubiquitin-like processing and binds to phosphatidylethanolamine on the surface of the autophagosome membrane to form the LC3II type. SQSTM1/p62 is a multifunctional ubiquitination-bound adaptor protein encoded by the SQSTM1 gene, which participates in the protein degradation processes of the ubiquitin proteasome system and the autophagy–lysosome system. When autophagy activity is weakened in the early stages of neurodegenerative disease, the p62 protein accumulates in the cytoplasm. p62 can form a complex with ubiquitinated proteins and LC3II proteins on the autophagosome membrane to complete the degradation process in the autophagolysosome. Lysosomal storage disorders are also often characterized by a severe neurodegenerative phenotype. ATRA has been successfully used to treat acute promyelocytic leukemia (APL), and its induced differentiation of the APL cell line NB4 involves the induction of autophagy [53]. In the present study, our results showed that Ellorarxine was able to significantly upregulate the autophagy levels of glial cells, neurons and microglia under induced stress conditions. Under the condition of serum-free medium-induced autophagy, the autophagy levels of C6 cells and SH-SY5Y cells treated with Ellorarxine were further significantly increased by 30% and 20%, respectively. Under peroxide and starvation stress, the expression of cytoplasmic p62 in C6 cells was significantly reduced by more than 50%. The upregulation of LC3BII and the downregulation of p62 suggest that Ellorarxine increases and regulates the level of cellular autophagy. This suggests that Ellorarxine may induce cellular autophagy, facilitate the clearance of protein aggregates, and function as an autophagy inducer for the potential treatment of neurodegenerative diseases.

In this study, we provided the first evidence for the RARβ selectivity of Ellorarxine, and, furthermore, reported evidence that Ellorarxine protected all three brain cell types, C6, SH-SY5Y and HMC3, from oxidative and inflammatory stress. Specifically, mitochondrial dysfunction and cell death induced by oxidative and inflammatory stress in human SH-SY5Y neurons and neuroinflammation in human HMC3 microglia could be ameliorated through Ellorarxine pretreatment and autophagy promoted in SH-SY5Y and C6 glial cells.

## 4. Materials and Methods

### 4.1. Cell Lines and Culture

C6 (rat glioma), HMC3 (human microglial clone 3), and SH-SY5Y (human neuroblastoma) cells were obtained from Durham University and cultured in Dulbecco’s modified Eagle’s medium (DMEM, Gibco, London, UK) supplemented with 10% fetal bovine serum (FBS, Gibco, London, UK) and 1% Penicillin–Streptomycin Solution (Pen-Strep, Lonza, Slough, UK) at 37 °C in a humidified 5% incubator (Table 1). The growth medium was changed every 2 days. When the culture reached 80% confluence, trypsin–EDTA was added and incubated for 3–5 min to make adherent cells detach. Triturated cells were seeded in a ratio of 1:2 into 24-well plates or T75 flasks for further growth.

### 4.2. Cell Differentiation

The cells were differentiated by adding retinoic acid (RA) to Dulbecco’s modified Eagle’s medium (DMEM, Gibco, London, UK) with 1% Penicillin–Streptomycin Solution (Pen-Strep, Lonza, Slough, UK) to a final concentration of 10 μM, 24 h after subculturing. The cultures were differentiated for 6 days. The medium was changed every 2 days. After being differentiated, cells were cultured under normal conditions for two days to eliminate the effects of RA. The differentiation of neurons was confirmed based on MAP2 labelling and neurite outgrowth [48]. Differentiated cultures were used for all the treatments mentioned hereafter [25].

### 4.3. Preparation of Ellorarxine

Ellorarxine (1mM in DMSO) was synthesized following Nevrargenics’ patent of DC645 [53] and was stored at −20 °C. The drug was prepared to a 1 μM stock solution using dH_2_O and was stored at 4 °C. The test concentration was 10 nM, determined based on pilot studies, which showed that 10 nM produced maximal genomic and non-genomic effects [54,55].

### 4.4. Pretreatments

After trypsinization, cells were plated (40,000 cells/mL) in 24-well plate chambers and left to grow for 24 h at 37 °C and 5% CO_2_ before being treated with 10% DMSO (Sham treatment) or 10 nM Ellorarxine for 4 h before being stressed.

Oxidative stress was induced in the cells using H_2_O_2_, with a concentration determined in our preliminary experiments to result in approximately 50% mitochondrial viability. Inflammation stress was induced in the cells using LPS, with a concentration determined in our preliminary experiments to result in substantial cytokine release. These procedures were carried out following the methodology described in our previous research [48].

Autophagy was induced by culturing cells with DMEM/F12 without serum for 24 h (starvation stress) before the Ellorarxine treatment.

### 4.5. Methyl Thiazolyl Diphenyl Tetrazolium Bromide (MTT) Assay [32]

A total of 50 μL of 5 mg/mL MTT (M2128, Merck Life Science UK Limited, Gillingham, Dorset, UK) (Table 1) was added to each well and left to incubate for 4 h at 37 °C and 5% CO_2_. Subsequently, the medium was removed and 200 μL DMSO was added to each well to dissolve the formazan crystals. Finally, 100 μL from each well was transferred to a 96-well tissue culture plate, and the absorbance was measured at 595 nm using a microplate reader.

### 4.6. Lactate Dehydrogenase (LDH) Release Assay [33]

The LDH release was measured using a CytoTox 96 kit (ADG1781, Promega, Southampton, UK) (Table 1). A total of 100 μL of the supernatant was extracted from each well and transferred to a 96-well tissue culture plate. A total of 100 μL of the cytotoxicity detection kit LDH solution was added to each well and incubated for 30 min in the dark at room temperature. The reaction was stopped by adding 50 μL of the stop solution. Subsequently, the optical density was measured at 490 nm. This assay was normalized by freezing the leftover plate, later thawing it, then pipetting the contents of each well into Eppendorf tubes, centrifuging those for 10 min for the cells to settle down, and then extracting 100 μL of the supernatant from each Eppendorf tube and following the same procedure as described above. This gave us an indication of the total amount of LDH and allowed for normalization.

### 4.7. Enzyme-Linked Immunosorbent Assay (ELISA) [34]

After 24 h had passed since stressing the cells, 100 μL of the supernatant was collected from each well and an ELISA was carried out using the Human IL-6 ELISA kit (ab178013, Abcam, Cambridge, UK) and Human TNF-α ELISA kit (ab46087) according to the manufacturer’s protocol (Table 1). The standard curve generated was used to calculate concentrations from the absorbance measurements.

### 4.8. Immunocytochemistry Staining [37,38]

Cells were plated at a density of 8000/mL in 6-well (35 mm) chambers onto 15 mm × 15 mm coverslips. After 24 h had passed since stressing the cells, immunocytochemistry staining was carried out using the VECTASTAIN Elite ABC Universal Kit (PK-6200), 2BScientific, Kirtlington, UK and ImmPACT DAB Substrate Kit, Peroxidase (SK-4105), 2BScientific, Kirtlington, UK, according to the manufacturer’s protocol. The primary antibodies for LC3B were diluted in a ratio of 1:200 (Invitrogen, PA146286, 5781 Van Allen Way, Carlsbad, CA, USA).

### 4.9. Immunofluorescence Staining

Cells were plated at a density of 8000/mL in 6-well (35 mm) chambers onto 15 mm × 15 mm coverslips. After 24 h had passed since treatment, the cells were fixed in 4% paraformaldehyde (PFA) for 10 min at room temperature. Cells were washed three times for 5 min with PBS and then blocked in PBS containing 1% bovine serum albumin, 1% fish skin gelatin and 0.3% Triton X-100 at room temperature for 1 h. Then, the cells were incubated with the primary antibodies for 1 h at room temperature. The primary antibodies were diluted for RARα in a ratio of 1:100 (Abcam, Cambridge, UK, ab275745), for RARβ in a ratio of 1:100 (Abcam, ab5792), for RARγ in a ratio of 1:100 (Abcam, Cambridge, UK, ab97569), for Cyp26B1 in a ratio of 1:200 (Abcam, Cambridge, UK, ab113236) and for Anti-SQSTM1/p62 in a ratio of 1:200 (Abcam, Cambridge, UK, ab240635) (Table 1). Cells were then washed three times for 5 min in PBS and incubated with secondary antibodies (Goat Anti-Mouse IgG H&L Alexa Fluor^®^ 488, 1:1000, Abcam, Cambridge, UK, ab150113) for 1.5 h at room temperature. Cells were then washed three times for 5 min with PBS and incubated with DAPI (1 μg/mL) for 5 min at room temperature to stain the DNA for nuclear localization. Fluorescent images were captured by using a Zeiss fluorescent microscope (Zeiss ApoTome, Cambridge, UK) [31].

### 4.10. Quantification and Statistical Analysis

The semi-quantitative analysis of immunofluorescence images and immunocytochemistry images was conducted using ImageJ [31,38].

The data were obtained from at least three independent experiments for each experimental condition. The data were expressed as the means ± the SD. *t*-tests (Figure 1, Figure 2 and Figure 6) and two-way ANOVA tests (Figure 3, Figure 4, Figure 5 and Figure 7) were used to analyze the differences between the two groups. *p* values < 0.05 were considered significant. All these analyses were performed using Graphpad Prism 8. Key statistical results for each panel in the figures are shown in the figure legends.

**Table 1 ijms-26-03551-t001:** Reagents and Resources.

Reagent or Resource	Source	Identifier
**Antibodies**		
Recombinant Anti-Retinoic Acid Receptor alpha antibody [EPR23871-271] (ab275745)	Abcam (Cambridge, UK)	https://www.abcam.com/products/primary-antibodies/retinoic-acid-receptor-alpha-antibody-epr23871-271-ab275745.html, accessed on 1 March 2025
Anti-Retinoic Acid Receptor beta antibody (ab5792)	Abcam (Cambridge, UK)	https://www.abcam.com/products/primary-antibodies/retinoic-acid-receptor-beta-antibody-ab5792.html, accessed on 1 March 2025
Anti-Retinoic Acid Receptor gamma antibody (ab97569)	Abcam (Cambridge, UK)	https://www.abcam.com/products/primary-antibodies/retinoic-acid-receptor-gamma-antibody-ab97569.html, accessed on 1 March 2025
Anti-Cyp26B1 antibody (ab113236)	Abcam (Cambridge, UK)	https://www.abcam.com/products/primary-antibodies/cyp26b1-antibody-ab113236.html, accessed on 1 March 2025
Goat Anti-Mouse IgG H&L (Alexa Fluor^®^ 488) (ab150113)	Abcam (Cambridge, UK)	https://www.abcam.com/products/secondary-antibodies/goat-mouse-igg-hl-alexa-fluor-488-ab150113.html, accessed on 1 March 2025
Anti-SQSTM1/p62 antibody [EPR23101-103]	Abcam (Cambridge, UK)	https://www.abcam.com/en-us/search?productSorting=relevance&resourceSorting=relevance&keywords=sqstm1-p62-antibody-2c11-bsa-and-azide-free-ab56416&utm_source=google&utm_medium=cpc&gad_source=1&gclid=CjwKCAjwktO_BhBrEiwAV70jXmFhWo8-FX1QCxX-PhJIrI8dVqZnbe-AAwAeTuxAXQ896rDmkyFkKRoCdYQQAvD_BwE&gclsrc=aw.ds&productcode=ab56416&view=quickview, accessed on March 2025
LC3B Polyclonal Antibody, Invitrogen™	Thermo-Fisher (Altrincham, Cheshire)	https://www.fishersci.com/shop/products/lc3b-polyclonal-antibody-invitrogen-2/PA146286?searchHijack=true&searchTerm=PA1-46286&searchType=RAPID&matchedCatNo=PA1-46286, accessed on 1 March 2025
**Chemicals, peptides and recombinant proteins**		
Phosphate-buffered saline (P5368-10pak)	Merck Life Science UK Limited. (Gillingham, Dorset, UK)	https://www.sigmaaldrich.com/GB/en/search/p5368-10pak?focus=products&page=1&perpage=30&sort=relevance&term=p5368-10pak&type=product, accessed on 1 March 2025
Triton X-100		
BSA		
Tween 20		
Thiazolyl Blue Tetrazolium Bromide	Merck Life Science UK Limited. (Gillingham, Dorset, UK)	https://www.sigmaaldrich.com/GB/en/search/m2128?focus=products&page=1&perpage=30&sort=relevance&term=m2128&type=product, accessed on 1 March 2025
H_2_O_2_		
LPS		
**Critical commercial assays**		
CytoTox 96^®^ Non-Radioactive Cytotoxicity Assay	Promega (Chilworth, Southampton, UK)	https://www.promega.co.uk/products/cell-health-assays/cell-viability-and-cytotoxicity-assays/cytotox-96-non_radioactive-cytotoxicity-assay/?catNum=G1780, accessed on 1 March 2025
Human IL-6 ELISA kit (ab178013)	Abcam (Cambridge, UK)	https://www.abcam.com/products/elisa/human-il-6-elisa-kit-ab178013.html, accessed on 1 March 2025
Human TNF alpha ELISA kit (ab46087)	Abcam (Cambridge, UK)	https://www.abcam.com/products/elisa/human-tnf-alpha-elisa-kit-ab46087.html, accessed on 1 March 2025
VECTASTAIN^®^ Elite^®^ ABC-HRP Kit (Peroxidase, Universal) (PK-6200)	VECTASTAIN (2BScientific, Kirtlington, UK)	https://vectorlabs.com/products/vectastain-elite-abc-hrp-kit-universal, accessed on 1 March 2025
ImmPACT^®^ DAB Substrate Kit, Peroxidase (HRP) (SK-4105)	VECTASTAIN (2BScientific, Kirtlington, UK)	https://vectorlabs.com/products/immpact-dab-hrp-substrate, accessed on 1 March 2025
**Experimental models: cell lines**		
Rat glioma C6 cells		
Human neuroblastoma SH-SY5Y cells		
Human microglia clone 3 HMC3 cells		
**Software and algorithms**		
ZEN software	Zeiss (Cambridge, UK)	https://www.zeiss.com/microscopy/en/products/software/zeiss-zen.html, accessed on 1 March 2025
Prism8	GraphPad, Prism, (Boston, MA, USA)	https://www.graphpad.com/, accessed on 1 March 2025
ImageJ	LOCI	https://imagej.net/, accessed on 1 March 2025

## 5. Patents

Whiting A, Valentine R, Chisholm DR, McCaffery P, Greig IR, Khatib T (2019) Synthetic retinoids for use in RAR activation. Patent No. GB1903242.4.

## Figures and Tables

**Figure 1 ijms-26-03551-f001:**
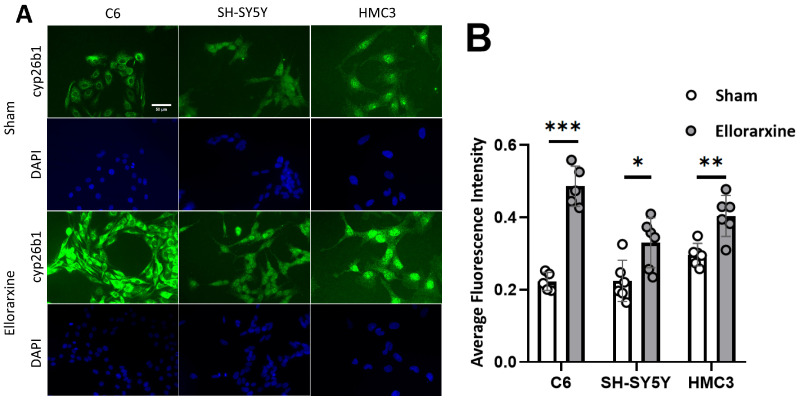
Ellorarxine upregulates expression of Cyp26b1. (**A**) Immunofluorescence staining of Cyp26b1 (green) and DAPI (blue) in C6, SH-SY5Y and HMC3 cells. Scale bar: 50 μm. (**B**) Average fluorescence intensity of Cyp26b1 in C6, SH-SY5Y and HMC3 cells, n = 6 per group. Data are presented as mean ± SD. * *p* < 0.05. ** *p* < 0.01. *** *p* < 0.001.

**Figure 2 ijms-26-03551-f002:**
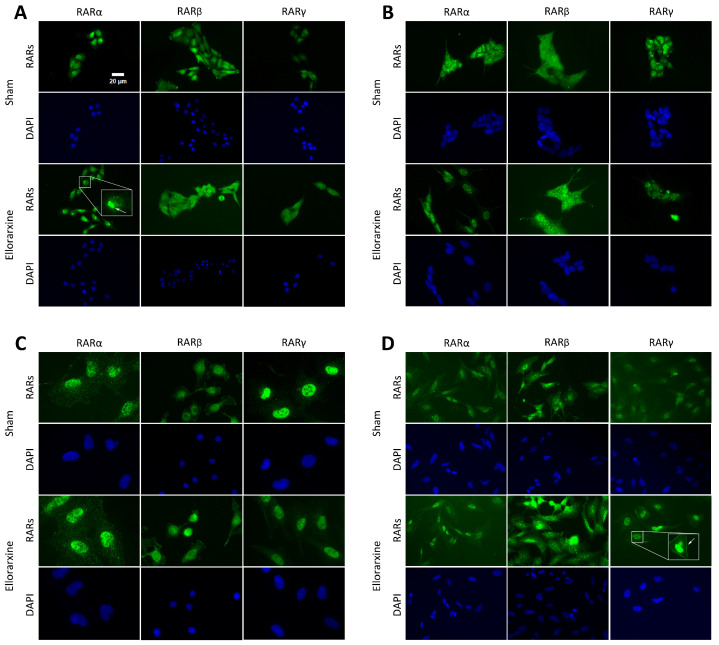
Ellorarxine selectively upregulates expression of RARβ. (**A**) Immunofluorescence staining of RARα, RARβ, RARγ (green) and DAPI (blue) in C6 cells. (**B**) Immunofluorescence staining of RARα, RARβ, RARγ (green) and DAPI (blue) in SH-SY5Y cells. (**C**) Immunofluorescence staining of RARα, RARβ, RARγ (green) and DAPI (blue) in HMC3 cells. (**D**) Immunofluorescence staining of RARα, RARβ, RARγ (green) and DAPI (blue) in differentiated SH-SY5Y cells. (**E**) Average fluorescence intensity of RARα, RARβ and RARγ in C6 cells, n = 4 per group. Data are presented as mean ± SD. (**F**) Average fluorescence intensity of RARα, RARβ and RARγ in SH-SY5Y cells, n = 4 per group. Data are presented as mean ± SD. (**G**) Average fluorescence intensity of RARα, RARβ and RARγ in HMC3 cells, n = 4 per group. Data are presented as mean ± SD. (**H**) Average fluorescence intensity of RARα, RARβ and RARγ in differentiated SH-SY5Y cells, n = 4 per group. Data are presented as mean ± SD. (**I**) Average fluorescence intensity of RARα in C6 cells. Shows average fluorescence intensity of individual pixels on diameter line of cell nucleus. n = 4 per group. Arrow indicates peak of expression. Data are presented as mean ± SD. (**J**) Average fluorescence intensity of RARγ in differentiated SH-SY5Y cells. Shows average fluorescence intensity of cell membrane. ** p* < 0.05, **** p<* 0.005. Scale bar: 20 μm.

**Figure 3 ijms-26-03551-f003:**
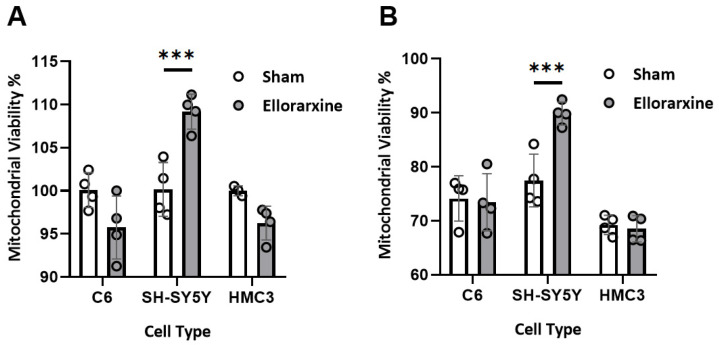
Ellorarxine pretreatment alleviates mitochondrial dysfunction. (**A**) Mitochondrial viability in C6, SH-SY5Y, and HMC3 cells treated with 10% DMSO (Sham) and 10 nM Ellorarxine. n = 4 per group. (**B**) Mitochondrial viability in C6, SH-SY5Y, and HMC3 cells pretreated with 10% DMSO (Sham) and 10 nM Ellorarxine under 100 µM H_2_O_2_ stress. n = 4 per group. Data are presented as mean ± SD. **** p* < 0.001.

**Figure 4 ijms-26-03551-f004:**
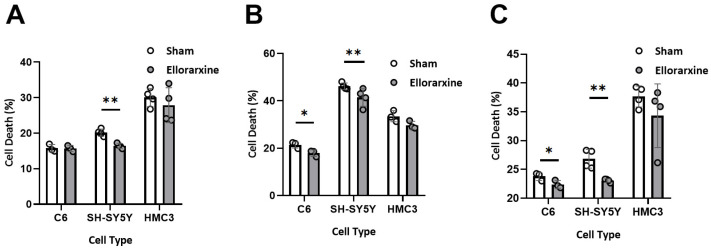
Ellorarxine pretreatment reduced cell death. (**A**) Cell death in C6, SH-SY5Y, and HMC3 cells treated with 10% DMSO (Sham) and 10 nM Ellorarxine. n = 4 per group. (**B**) Cell death in C6, SH-SY5Y, and HMC3 cells pretreated with 10% DMSO (Sham) and 10 nM Ellorarxine under 200 mM H_2_O_2_ stress. n = 4 per group. (**C**) Cell death in C6, SH-SY5Y, and HMC3 cells pretreated with 10% DMSO (Sham) and 10 nM Ellorarxine under 10 μg/mL LPS stress. n = 4 per group. Data are presented as mean ± SD. * *p* < 0.05, ** *p* < 0.01.

**Figure 5 ijms-26-03551-f005:**
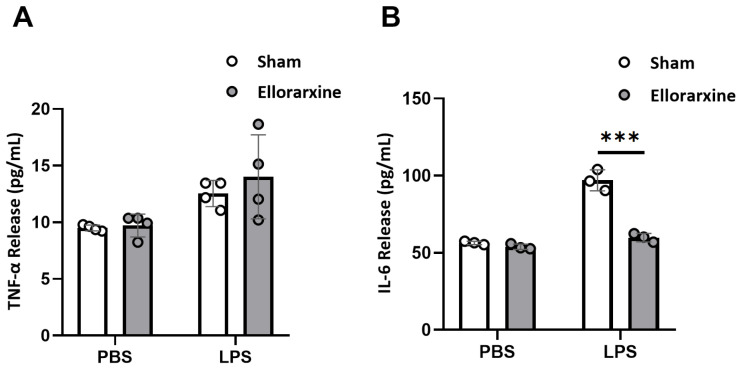
Ellorarxine pretreatment selectively modulated microglia inflammatory cytokine release. (**A**) TNF-α release under LPS stress with 10% DMSO (Sham) and 10 nM Ellorarxine. n = 4 per group. (**B**) IL-6 release under LPS stress with 10% DMSO (Sham) and 10 nM Ellorarxine. n = 3 per group. Data are presented as mean ± SD. *** *p* < 0.001.

**Figure 6 ijms-26-03551-f006:**
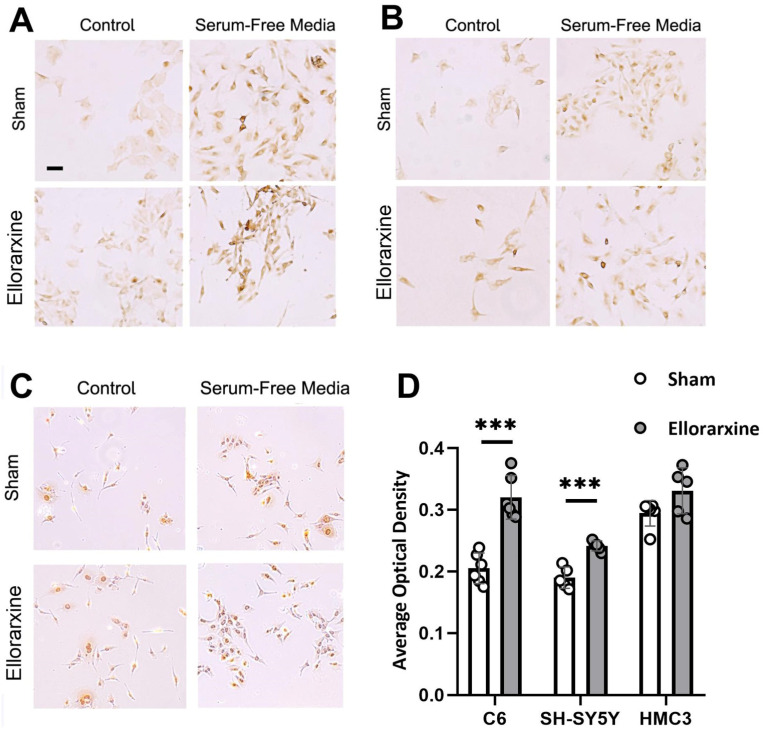
Ellorarxine treatment regulated cellular autophagy. (**A**) LC3BII immunocytochemical staining of C6 cells. (**B**) LC3BII immunocytochemical staining of SH-SY5Y cells. (**C**) LC3BII immunocytochemical staining of HMC3 cells. (**D**) Average Optical Density in serum-free media condition with 10% DMSO (Sham) and 10 nM Ellorarxine. n = 6 per group. Data are presented as mean ± SD. *** *p* < 0.001. Scale bar: 20 μm.

**Figure 7 ijms-26-03551-f007:**
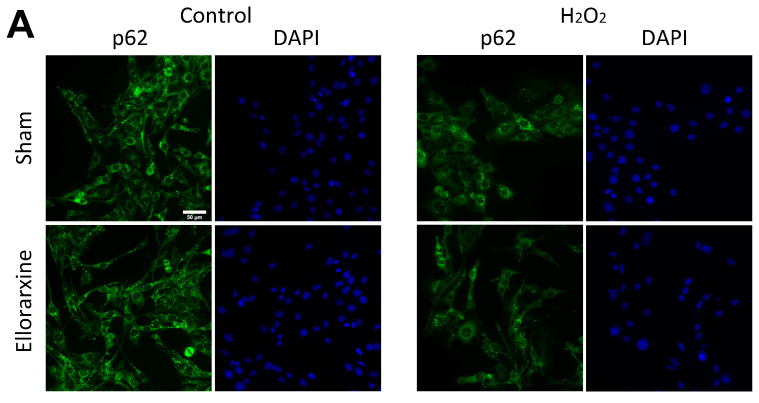
(**A**) Immunofluorescence staining of p62 and DAPI in C6 cells. Scale bar: 50 μm. (**B**) Immunofluorescence staining of p62 and DAPI in SH-SY5Y cells. Scale bar: 10 μm. (**C**) Immunofluorescence staining of p62 and DAPI in HMC3 cells. Scale bar: 50 μm. (**D**) Average fluorescence intensity of p62 under H_2_O_2_ stress in C6, SH-SY5Y and HMC3 cells, n = 8 per group. (**E**) Average fluorescence intensity of p62 under unstressed conditions, in C6, SH-SY5Y and HMC3 cells, n = 8 per group. Data are presented as mean ± SD. *** *p* < 0.001.

## Data Availability

The data are contained within the article.

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
