# Peer review of "A Comparative Study of a Potent CNS-Permeable RARβ-Modulator, Ellorarxine, in Neurons, Glia and Microglia Cells In Vitro"

_ijms, 2025, doi:10.3390/ijms26083551_

Round 1

Reviewer 1 Report

Comments and Suggestions for Authors

In this manuscript, the authors examined the effects of Ellorarxine on neurodegeneration using three main neural cell types, neurons, astrocytes, and microglia, in vitro. They demonstrated that Ellorarxine mitigates detrimental phenotypes in these cells exposed to adverse conditions, including oxidative stress and neuroinflammation. The mechanisms underlying these protective effects were explored, highlighting a reduction in inflammatory cytokine release from microglia and an increase in autophagy across all three cell types, suggesting that Ellorarxine promotes cellular resilience under stress.

While the rationale for using SH-SY5Y and HMC3 cell lines to model neurodegenerative conditions was explained, the selection of C6 cells as an astrocyte model requires further clarification. Given that C6 cells originate from a glioma background, have additional experiments been conducted to determine whether primary or immortalized astrocytes exhibit similar responses in these assays? Providing such validation would strengthen the translational relevance of the findings.

Additionally, since Ellorarxine functions as an RAR agonist and modulates its activity, what is the proposed mechanism by which it increases RAR expression? Could this be indicative of a positive feedback loop in which Ellorarxine binding to RAR further promotes its own expression? Instead of relying solely on RAR expression levels, have you examined the expression of well-established RAR target genes, such as Cyp26a1? Measuring the expression of these target genes would provide a more direct and functionally relevant readout of RAR activity.

Since ligand-bound RAR activation typically involves its translocation from the cytosol to the nucleus to regulate gene expression, has immunostaining been performed to confirm increased nuclear localization of RAR following Ellorarxine treatment? The manuscript also mentions that Ellorarxine influences the topology of RARα and RARγ—please include quantitative data to support this claim.

Finally, for most of the data analyses, a two-way ANOVA would be more appropriate than simple t-tests for comparing two groups. This approach would provide a more accurate assessment of both the effects of stress and Ellorarxine on mitochondrial activity, cell death, autophagy, and related outcomes. Additionally, it would offer a clearer perspective on the extent to which Ellorarxine rescued stressed cells and how closely this rescue approximates unstressed, control cells.

The limitations of the study stem from the use of artificial cell lines and not primary cells. The study is mostly observational and therefore, findings that Ellorarxine is neuroprotective has not been rigorously evaluated. Some of the suggested experiments could improve the clarity and rigor of the data interpretation. 

Comments on the Quality of English Language

The paper is mostly written properly. Some editing for grammar and sentence clarity is needed. 

Author Response

In this manuscript, the authors examined the effects of Ellorarxine on neurodegeneration using three main neural cell types, neurons, astrocytes, and microglia, in vitro. They demonstrated that Ellorarxine mitigates detrimental phenotypes in these cells exposed to adverse conditions, including oxidative stress and neuroinflammation. The mechanisms underlying these protective effects were explored, highlighting a reduction in inflammatory cytokine release from microglia and an increase in autophagy across all three cell types, suggesting that Ellorarxine promotes cellular resilience under stress.

While the rationale for using SH-SY5Y and HMC3 cell lines to model neurodegenerative conditions was explained, the selection of C6 cells as an astrocyte model requires further clarification. Given that C6 cells originate from a glioma background, have additional experiments been conducted to determine whether primary or immortalized astrocytes exhibit similar responses in these assays? Providing such validation would strengthen the translational relevance of the findings.

All three cell types used in this study are widely used clonal cell lines. In fact. the C6 is the most popular and widely neuroblastoma line and are considered to be of astrocytic lineage, meaning it originates from astrocytes.

Additionally, since Ellorarxine functions as an RAR agonist and modulates its activity, what is the proposed mechanism by which it increases RAR expression? Could this be indicative of a positive feedback loop in which Ellorarxine binding to RAR further promotes its own expression? Instead of relying solely on RAR expression levels, have you examined the expression of well-established RAR target genes, such as Cyp26a1? Measuring the expression of these target genes would provide a more direct and functionally relevant readout of RAR activity.

Retinoids has well known effects on endogenous transcriptional regulation in the SH-SY5Y cell line used for neurite outgrowth and non-genomic activity assays, two genes with well-defined RAREs in their promoters were investigated: RARβ and CYP26A. 1EC23, AH61, EC23Al, TTNN and JBGG179 as well as ATRA induce both CYP26A1 and RARβ experession.

Khatib, T., Marini, P., Nunna, S. et al. Genomic and non-genomic pathways are both crucial for peak induction of neurite outgrowth by retinoids. Cell Commun Signal 17, 40 (2019). https://doi.org/10.1186/s12964-019-0352-4

Since ligand-bound RAR activation typically involves its translocation from the cytosol to the nucleus to regulate gene expression, has immunostaining been performed to confirm increased nuclear localization of RAR following Ellorarxine treatment? The manuscript also mentions that Ellorarxine influences the topology of RARα and RARγ—please include quantitative data to support this claim.

This was done, Figure 2I. and J.

Finally, for most of the data analyses, a two-way ANOVA would be more appropriate than simple t-tests for comparing two groups. This approach would provide a more accurate assessment of both the effects of stress and Ellorarxine on mitochondrial activity, cell death, autophagy, and related outcomes. Additionally, it would offer a clearer perspective on the extent to which Ellorarxine rescued stressed cells and how closely this rescue approximates unstressed, control cells.

We agree with the reviewer for some of the figures. t-tests (Fig.1,2,6) and two-way ANOVA tests (Fig.3,4,5,7) were used to analyze differences between the two groups.  p values <0.05 are considered significant. 

The limitations of the study stem from the use of artificial cell lines and not primary cells. The study is mostly observational and therefore, findings that Ellorarxine is neuroprotective has not been rigorously evaluated. Some of the suggested experiments could improve the clarity and rigor of the data interpretation. 

We agree with reviewer that these are cell lines and the first step in the in vitro mechanistic study. This study complement a recently published study providing further evidence to support the neuroprotective, neuroplasticity and neurorepair effects of Ellorarxine (escudier et al., 2024)

Escudier O, Zhang Y, Whiting A, Chazot P. Evaluation of a Synthetic Retinoid, Ellorarxine, in the NSC-34 Cell Model of Motor Neuron Disease. Int J Mol Sci. 2024 Sep 10;25(18):9764. doi: 10.3390/ijms25189764. PMID: 39337251; PMCID: PMC11431449.

Reviewer 2 Report

Comments and Suggestions for Authors

The study showed that Ellorarxine upregulates Cyp26b1 and RARbeta, has a protective effect on SH-SY5Y mitochondria, reduces C and SH-SY5Y cell mortality, release of IL6 from HMC3 cells and increases cellular autophagy. The results are interesting, but some content requires explanation.

  • Full name of GFAP (line 71), SOD (line 229)
  • CYP26B1 should be Cyp26b1 (line 55).
  • The sentence: "RARs are... and RARs[30]" (lines 80-81) should be transferred into the Introduction.
  • How was mitochondrial dysfunction (mitochondrial viability) assessed? Which of the research methods is reported for these studies? How was it known that 100uM H202 caused mitochondrial dysfunction (add reference) and generated oxidative stress (some marker of oxidative stress?).
  • Why is cytokine release only shown in HMC3 cells?
  • The sentences describing the validity of LC3 determination (paragraph 2.5, lines 156-167) should have been moved to the discussion. They do not directly describe the results.
  • TNF alpha has been called TNF for several years. If the name of the reagent is given, it should be given as it is (e.g. TNF-alpha...), but if this cytokine is discussed, the name consistent with the nomenclature should be used: TNF.
  • Character errors, e.g. supple-mented (line 262), car-ried (line 304), accord-ing (line 305). Please check the entire text carefully.
  • The specifications of the cell lines used in the study should be provided (point 4.2).
  • How do you know that the cells have differentiated? Were there any signs of differentiation observed? (line 272).
  • Paragraph 4.10 is missing.
  • What is the conclusion that ellorarxine increases the level of cellular autophagy

Author Response

The study showed that Ellorarxine upregulates Cyp26b1 and RARbeta, has a protective effect on SH-SY5Y mitochondria, reduces C and SH-SY5Y cell mortality, release of IL6 from HMC3 cells and increases cellular autophagy. The results are interesting, but some content requires explanation.

  • Full name of GFAP (line 71), SOD (line 229)

This was done

  • CYP26B1 should be Cyp26b1 (line 55).

This was done

  • The sentence: "RARs are... and RARs[30]" (lines 80-81) should be transferred into the Introduction.

This was done

  • How was mitochondrial dysfunction (mitochondrial viability) assessed? Which of the research methods is reported for these studies? How was it known that 100uM H202 caused mitochondrial dysfunction (add reference) and generated oxidative stress (some marker of oxidative stress?).

Mitochondrial dysfuction was assayed using a classic well used MTT assay. We selected this peroxide concentration based on our previous dose response assays, to elicit approx. 50% mitochondrial viability (oxidative stress). MTT assay uses a colorimetric method to measure cellular metabolic activity, which is an indicator of cell viability. Enzymatic reduction of 3-[4,5-dimethylthiazole-2-yl]-2,5-diphenyltetrazolium bromide (MTT) to MTT-formazan is catalyzed by mitochondrial succinate dehydrogenase. Hence, the MTT assay is dependent on mitochondrial respiration and indirectly serves to assess the cellular energy capacity of a cell.

Escudier O, Zhang Y, Whiting A, Chazot P. Evaluation of a Synthetic Retinoid, Ellorarxine, in the NSC-34 Cell Model of Motor Neuron Disease. Int J Mol Sci. 2024 Sep 10;25(18):9764. doi: 10.3390/ijms25189764. PMID: 39337251; PMCID: PMC11431449.

  • Why is cytokine release only shown in HMC3 cells?

HMC3 microglia cells were used predominately, as neither of the other cell types elicited TNFa or IL-6 release, upon LPS activation.

  • The sentences describing the validity of LC3 determination (paragraph 2.5, lines 156-167) should have been moved to the discussion. They do not directly describe the results.

Done

  • TNF alpha has been called TNF for several years. If the name of the reagent is given, it should be given as it is (e.g. TNF-alpha...), but if this cytokine is discussed, the name consistent with the nomenclature should be used: TNF.

Done

  • Character errors, e.g. supple-mented (line 262), car-ried (line 304), accord-ing (line 305). Please check the entire text carefully.

Done

  • The specifications of the cell lines used in the study should be provided (point 4.2).

Done

  • How do you know that the cells have differentiated? Were there any signs of differentiation observed? (line 272).

Differentiation of neurons was confirmed by MAP2 labelling and/or neurite outgrowth, as reported in “Escudier O, Zhang Y, Whiting A, Chazot P. Evaluation of a Synthetic Retinoid, Ellorarxine, in the NSC-34 Cell Model of Motor Neuron Disease. Int J Mol Sci. 2024 Sep 10;25(18):9764. doi: 10.3390/ijms25189764. PMID: 39337251; PMCID: PMC11431449”

  • Paragraph 4.10 is missing.

Corrected

  • What is the conclusion that ellorarxine increases the level of cellular autophagy

This has been expanded to explain this conclusion